# Characteristics and Clinical Application of Extracellular Vesicle-Derived DNA

**DOI:** 10.3390/cancers13153827

**Published:** 2021-07-29

**Authors:** Jae Young Hur, Kye Young Lee

**Affiliations:** 1Precision Medicine Lung Cancer Center, Konkuk University Medical Center, Seoul 05030, Korea; 20160475@kuh.ac.kr; 2Department of Pathology, Konkuk University Medical Center, Seoul 05030, Korea; 3Department of Pulmonary Medicine, Konkuk University School of Medicine, Seoul 05030, Korea

**Keywords:** extracellular vesicle, exosome, microvesicle, EV DNA, liquid biopsy, next-generation sequencing, methylation, gene transfer

## Abstract

**Simple Summary:**

We reviewed characterization and the role of DNAs derived from extracellular vesicles focusing on its use for identifying biomarkers. Extracellular vesicles contain double-stranded genomic DNA reflecting the mutational status and methylation profile of the parental tumor cells. Many studies demonstrated higher stability, sensitivity, and specificity of extracellular vesicle DNAs in comparison to cell-free DNAs, demonstrating a high potential for clinical application as a source for liquid biopsy. Moreover, the horizontally transfer ability of extracellular vesicle DNAs could be utilized in therapeutics.

**Abstract:**

Extracellular vesicles (EVs) carry RNA, proteins, lipids, and diverse biomolecules for intercellular communication. Recent studies have reported that EVs contain double-stranded DNA (dsDNA) and oncogenic mutant DNA. The advantage of EV-derived DNA (EV DNA) over cell-free DNA (cfDNA) is the stability achieved through the encapsulation in the lipid bilayer of EVs, which protects EV DNA from degradation by external factors. The existence of DNA and its stability make EVs a useful source of biomarkers. However, fundamental research on EV DNA remains limited, and many aspects of EV DNA are poorly understood. This review examines the known characteristics of EV DNA, biogenesis of DNA-containing EVs, methylation, and next-generation sequencing (NGS) analysis using EV DNA for biomarker detection. On the basis of this knowledge, this review explores how EV DNA can be incorporated into diagnosis and prognosis in clinical settings, as well as gene transfer of EV DNA and its therapeutic potential.

## 1. Introduction

In light of recent developments in targeted therapies [1], immunotherapies [2], and precision medicine [3], the importance of liquid biopsy for detecting cancer DNA and biomarkers has become obvious [4]. Liquid biopsy can utilize all fluids produced by humans, such as blood [5], urine [6], saliva [7], cerebrospinal fluid [8], ascites [9], and pleural effusion [10]. Currently, blood biopsy is the most frequently utilized diagnostic method for almost all cancers [11]; however, sensitivity can be increased by using specific liquid samples that are related to specific cancers [12], for example, the urine for bladder cancer [13], saliva for head and neck cancer [14], and bronchoalveolar lavage fluid (BALF) for lung cancers [15]. For liquid biopsy, the cell-free nucleic acids (cfNA), circulating tumor cells (CTCs), and extracellular vesicles (EVs) that exist in these specimens are isolated using high-tech equipment, and the isolated materials are further analyzed using next-generation sequencing (NGS), real-time PCR, digital PCR, and/or bioinformatics (BI) for the screening and early detection of cancer [16,17]. Furthermore, the information can be helpful in deciding a targeted therapy as a companion diagnostic and in the monitoring of treatment response, drug resistance, recurrence, and metastasis [18,19,20,21].

EVs, a diverse population of biological particles with sizes ranging from approximately 30 to 1000 nm, emerged into the spotlight as an important source for liquid biopsy [22,23,24,25]. The characteristics of EVs include the representation of their original host through their cargo, showing their potential as biomarkers [15,26], and through the transfer of cellular components between cells [27,28]. Depending on their origin, biogenesis, and size, EVs are categorized into exosomes, microvesicles, microparticles, and apoptotic bodies [29,30]. However, the term EV is more commonly used in a broader sense, as isolating one specific subcategory remains technically difficult [31].

Early studies have found that EVs are more abundantly shed by cancer cells than normal cells [32]. The same pattern was observed in the plasma of ovarian cancer [33] and non-small-cell lung cancer (NSCLC) patients [34]. EVs carry RNA, DNA, proteins, lipids, and diverse bioactive materials [29]. Initially, studies were predominantly focused on EV RNAs; however, recently, EV-derived DNA (EV DNA) has garnered attention. EV DNAs largely consist of large genomic DNA (gDNA) and tumor-specific oncogenic mutant DNA, unlike fragmented cell-free DNA (cfDNA) [15,35,36]. Plasma cfDNA has been the go-to biomarker for diagnosis and prognosis by liquid biopsy because of its easy obtainability; however, the low sensitivity of this approach, because of the short half-life of plasma cfDNA, poses a challenge in its application [37,38]. In contrast, the structural stability of EVs renders EV DNA a more ideal subject than free-floating cfDNA, as the lipid bilayer of EVs protects EV DNA from degradation by external factors [39]. Recent publications in NGS studies have demonstrated that EV DNA can serve as a good cancer biomarker [40,41]. This review summarizes the current state of understanding of the traits of EV DNA, the DNA loading mechanism, and the application of EV DNA to NGS.

## 2. History of Extracellular Vesicles

Before the first publication that identified and used the term microvesicles in 1975 by Dalton AJ [42], the existence of EVs and their functions were recognized early in several studies, including studies on thromboplastic protein [43], platelet dust [44], and globules [45]. The description and properties of EVs were determined primarily using ultracentrifugation, electron microscopy (EM), and functional studies. Chargaff E. and West R. [43], in 1946, showed that high-speed centrifugation at 31,000× *g* for 150 min significantly extended the clotting time of the supernatant. Wolf P [44] visualized small vesicles originating from platelets and termed them “platelet dust” using electron microscopy, while Bonucci E [45] observed the calcifying property of EVs in the bone matrix.

By the 1980s, the shedding of EVs by tumor cells [46] and EV formation during reticulocyte maturation were recognized, and these vesicles were named exosomes [47,48], advancing the understanding of EV release. Researchers began identifying the contents and functions of EVs, beginning with the study by Ceccarini Met al. [49] in the late 1980s, which identified RNA-containing EVs, and a study in the 1990s that demonstrated immune cells secreting antigen-presenting EVs, which implied a transfer of information between cells [50].

In the 2000s, studies reported that EVs containing RNAs such as mRNA and microRNA have the ability to be transferred from cell to cell [51,52]. The identification of DNA associated with EVs and their possible functions began relatively early, with reports on EV and DNA binding in 1979 [53], resistance of the EV–DNA complex to DNase, and transfer of DNA via EVs in bacteria in 1982 [54]. Subsequently, through studies that showed the ability of EVs to transport their DNA cargo into host cells [55,56], others have reported that double-stranded DNA (dsDNA) in exosomes can be used as a biomarker in cancer detection [26,57].

Following the identification of EVs as carriers of DNA, RNA, and protein, the literature on EVs expanded substantially, with considerable attention given to the protein and RNA cargos. More recently, EV DNA, another important EV biomolecule, has emerged as a relevant and valuable material for cancer biology.

## 3. Characteristics of Extracellular Vesicle-Derived DNA

EVs exist in various forms, such as single vesicles, double vesicles, double-membrane vesicles, and multilayer vesicles [58,59]. EV DNAs exist in both single strand [60] and double-strand forms [61,62], along with the nuclear protein histone [63], enclosed in EVs (Figure 1). The presence of dsDNA in single vesicles and multilayer vesicles was observed using immuno-EM (Figure 2). Genomic DNA, mitochondrial DNA (mtDNA), and plasmid DNA have all been identified inside exosomes, microvesicles, and apoptotic bodies [63,64,65,66,67,68]. Apart from DNA enclosed within EV, it can be attached to the outer surface of EV, or both [35,63,64,69] (Figure 1). EV DNA can be detected in almost all body fluids, including the blood [57], urine [70], saliva [71], pleural effusion [10], BALF [15], ascites [72], and gastric juice [73] (Figure 3). The presence of dsDNA in EVs is well established; however, some reports have suggested that exosomes do not carry DNA [74]. This inconsistency in the results on the presence and absence of DNA can be attributed to the preparation method and size of the isolated EVs. If the isolation method is too rigorous, it may cause the loss of DNA-containing vesicles, leading to low DNA detection [75]. DNA is unlikely to exist inside an EV in the naked form; it exists in a nucleosome or supercoiled form, which would enable the packaging of large dsDNA into EVs such as oncosomes. In addition, considering that the size of the nucleosome is 11 nm [76], it is reasonable to assume that long dsDNA would likely not be present in EVs smaller than 50 nm in diameter. Several studies have utilized immuno-EM with anti-dsDNA to evaluate EVs isolated using size-based methods, and have identified them as exosomes [26,77]; basing the identification only on size (smaller than 200 nm) may not be correct. In a 2017 study by Takahashi et al. [78], DNA was observed in intraluminal vesicles (ILVs) inside multivesicular endosomes (MVEs) and not exosomes. Therefore, the presence of DNA in excreted exosomes remains controversial and will continue to be controversial until the development of a method for isolating pure exosomes or microvesicles.

Nonetheless, it is clear that DNA extracted from all categories of EVs is the latest and most promising biomarker for identifying tumor presence and complexity [79,80]. The size of dsDNA found in EVs ranges from ~100 bp to ~20 kbp [79], which can represent the entire genome and reflect the mutational status of tumor parental cells [15,26,81]. The EV nucleic acid (EV NA) population includes DNA and RNA of mutant or wild-type, and from this population, the target biomarker EV NA is detected more efficiently [82]. In addition, not using DNase in the DNA extraction process would increase the overall detection of mutant DNA, as this would include mutants from EV DNA and residual cfDNA in the sample [83,84,85].

Different studies and research teams have used various methods of EV isolation and DNA extraction that could impact DNA extraction efficiency. According to one study, adding polyethylene glycol improved general EV DNA isolation [86]. Additionally, the development of a microfluidic platform demonstrated that a new method can be used to successfully isolate EV DNA and monitor the residual or recurrent tumor presence in pancreatic cancer [87].

## 4. Extracellular Vesicle-Derived DNA Stability and Mutant Detection

EVs have been proven to increase the stability of their contents by protecting them from digestive enzymes and other biological fluids until the contents reach their target; hence, they play a role in enhancing the availability of bioactive compounds. [88,89]. Osteikoetxea X et al. [90] showed that EVs are resistant to detergent lysis owing to a liquid-ordered phase membrane. The stability of EVs extends beyond the isolation process, as they are in their optimal state even after isolation, as demonstrated by several studies. Isolated EVs can be safely stored for up to one year at temperatures below −80 °C with no coagulation [91,92] and up to 3 months at −0 °C [93]. In addition, almost all types of exosomes contained in biofluids can be stored for up to 5 days at 4 °C in a glass bottle [91]. Kumeda et al. [94] showed that the integrity of isolated exosomes was stable for up to 20 months when stored at 4 °C and 28 days when stored at 4 °C as whole saliva.

Moreover, the EV DNA from the serum remained stable for 1 week at 4 °C, 1 day at room temperature, and even after repeated freeze−thaw cycles [39]. In contrast, free-floating and circulating DNAs in body fluids, cfDNAs, undergo non-specific degradation [37,38]. The stability and abundance of EV DNA make it a good source for highly sensitive detection of DNA mutations by liquid biopsy [10,15,95,96,97]. *EGFR* genotyping using EV DNA derived from pleural effusion in lung cancer is a good example. Compared with cfDNA liquid biopsy and tissue genotyping, EV DNA genotyping resulted in 100% agreement in EGFR-tyrosine kinase inhibitor (TKI)-naïve patients. When detecting a biomarker *EGFR* mutation for EGFR-TKI resistance, T790M, using EV DNA even surpassed cell blocks or cfDNA in detection sensitivity [10]. In another study, the sensitivity and specificity of BALF EV-based *EGFR* genotyping were high and showed an even better mutation detection rate than tissue- or cytology-based typing of patients with lung cancer [98].

Collectively, these results show the stability and sensitivity of EV DNA-based genotyping and demonstrate that the highly promising liquid biopsy method is particularly efficient for patients that require repeated diagnosis throughout the disease progression. However, one must not forget that although EVs and their DNAs are stable, the structural and physicochemical properties could change because of several external factors such as pressure, freeze−thaw cycles, nature of the solvent, and storage duration [88].

## 5. Methylation of Extracellular Vesicle-Derived DNA

Apart from tumor-specific changes in its sequence, tumor DNA exhibits distinctive epigenetic marks and changes in DNA methylation [99]. Analysis of specific patterns of DNA methylation is attracting attention as a potential biomarker for the detection and diagnosis of diseases such as cancer [100,101]. Several studies have evaluated the methylation of EV DNA and demonstrated similarities in methylation profiles between gDNA and EV DNA in murine melanoma cells [26]; the gastric juice in patients with gastric cancer [73,102]; and the serum in patients with breast cancer and melanoma [103], metastatic castration-resistant prostate cancer [104], diffuse large B-cell lymphoma [105], and glioblastoma [106] (Table 1). These studies have revealed differences in the methylation of cancer cells and normal cells, and suggested methylation as a biomarker. Specifically, studies in patients with gastric cancer demonstrated methylation in *SOX17* [73] and *BARHL2* with a high sensitivity and specificity [102]. EV DNA purified from the serum of patients with breast cancer and melanoma cancer was hypomethylated compared with that from normal patients [103]. In addition, the significant genes in metastatic castration-resistant prostate cancer, *GSTP1*, *RASSF1A*, and *SLFN11*, were identified to be DNA methylated in both CTCs and exosomes [104]. In the case of diffuse large B-cell lymphoma, the promoter regions of *CDKN2A* and *CDKN2B* were determined to be methylated in both plasma exosomes and primary tumor tissue [105]. A study on glioblastoma demonstrated that EVs with an origin close to their corresponding cells are more likely to correctly identify the methylation class of the parental cells and original tumors [106].

Gingivitis-related methylation has been evaluated using ultracentrifugation and size exclusion chromatography for EV DNA derived from the saliva; no significant difference was observed between the healthy and gingivitis samples [71]. In another study on gingivitis and periodontitis, periodontitis-associated EV DNA exhibited significantly increased global 5-methylcytosine (5mC) and *N*6-methyl-2’-deoxyadenosine (m6dA) modification in the DNA, while the gDNA showed no difference between the gingivitis and normal samples [107].

These findings suggest that the methylation analysis of EV DNA can serve as a useful biomarker for the detection of various diseases, especially in the diagnosis of cancer. However, the source of EV DNA, sample collection, DNA extraction method, and the type of disease appear to affect methylation detection.

## 6. Mechanism of DNA Loading onto EV and Transfer of EV-Derived DNA

Several functional reasons have been suggested for the loading of DNA onto EVs. For instance, cells can excrete harmful cytoplasmic DNA using EVs containing chromosome fragments to maintain cellular homeostasis and prevent viral hijacking of the cellular machinery [78]. In normal cells, gDNA is mainly confined to the cell nucleus, and gDNA generally does not interact with the cytoplasmic multivesicular bodies (MVB) that produce exosomes [78]. Conversely, EVs contain gDNA and nuclear proteins [72], as well as contents associated with cell senescence and stimulation of the STING inflammatory pathway [78]. The inhibition of exosome secretion results in the accumulation of nuclear DNA in the cytoplasm, thereby activating the cytoplasmic DNA-sensing machinery [78]. The detection of gDNA predominantly in EVs derived from cancer cells rather than healthy cells could be caused by the loading of nuclear components, including micronuclei, into EVs [72]. Although this phenomenon is relatively well known, the mechanisms remain poorly understood.

To confirm the presence and quantify the amount of gDNA in exosomes, tetraspanins are most often used as protein markers for sorting out exosomes, and dsDNA-binding dye is used for quantifying the percentage of DNA-positive exosomes [72]. A previous study illustrated the involvement of micronuclei—extra-nuclear bodies that contain damaged chromosome fragments or whole chromosomes generated when the cell nucleus fails to properly segregate nuclear material. This result shows that micronucleus-originated DNA are loaded into exosomes through a CD63-mediated DNA shuttle. They knocked down CD63 in ovarian cancer cells, which prevented the loading of micronucleus-originated DNA into exosomes [72]. An earlier study suggested another pathway involving the depletion of the nuclear envelope protein emerin, a well-known feature of cancer, which causes nuclear shape instability and shedding of nuclear-derived EVs that contain genomic material as a possible EV DNA-loading mechanism [108] (Figure 4). In addition, the biogenesis of oncogenes containing EVs in brain tumor cells may be independent of Rb/TP53 and acid sphingomyelinase (ASMase) pathways [62], both of which are vesiculation pathways of exosomes [109] and microvesicle biogenesis [110], respectively.

An additional possibility is the loading of DNA into tumor-derived exosomes in a manner similar to the mechanism of early vital NETosis, a form of neutrophil-specific cell death characterized by the expulsion of DNA that forms web-like structures referred to as neutrophil extracellular traps (NETs) against bacterial infection [111]. The mechanisms of dsDNA loading onto EVs and the secretion of EVs appear to be similar to the loading and secretion of virus dsDNA via EVs [112,113].

One of the functions of EVs is the transmission of signals from donors, providing a mechanism that can directly alter signaling in the recipient cell, leading to the exchange of complex information [114,115]. Although EVs vary in size, composition, and abundance, they often contain functional transmembrane proteins, lipids, mRNAs, and microRNAs [30,116]. Several reports have raised doubts about the direct loading of nucleic acids into EVs and their functionality when taken up by recipient cells. These studies showed that siRNA transfection into EVs by electroporation induces the formation of insoluble siRNA aggregates [117], and EVs loaded with plasmid DNAs by electroporation delivered DNA to recipient cells; however, they were not functionally active [118]. In addition, testing with human embryonic kidney cell-derived exosomes, transfected mRNAs, siRNAs, and plasmid DNA failed to induce or downregulate the protein expression in recipient cells [119].

Conversely, many recent studies have shown EV-mediated intercellular communication by horizontal gene transfer (HGT) with pre-existing mutations, demonstrating that EV DNAs are functional when transferred to recipient cells [120] (Table 2). Recipient cells receive genes through the fusion or internalization of EVs, and the biomaterial contents of EVs are transported into the nuclear compartment through nuclear envelope invagination-associated late endosomes [121,122]. This would require interactions between tumor cells and the microenvironment, including the composite of heterogeneous cells that populate the tumor. According to previous reports, other types of EVs can mediate ssDNA and dsDNA transfers. Most intercellular transfer of oncogenic DNA has been attributed to the uptake of exosomes, microvesicles, and apoptotic bodies [60,62,123]. For example, integrated viral genes are horizontally transferred by the uptake of DNA from apoptotic bodies [124]. Tumor DNA is horizontally transferred by the uptake of apoptotic bodies. The phagocytosis of apoptotic bodies derived from oncogenic *H-ras*- and human *c-myc*-transfected rat fibroblasts resulted in the development of a tumorigenic phenotype in mouse cells undergoing malignant transformation. The DNA transfer was confirmed by fluorescence in situ hybridization analysis, which showed the presence of donor DNA in the recipient cells [123]. A study demonstrated that donor cell EV gDNA can be transferred to recipient cells, and it can increase the mRNA and protein expression and even change function. Moreover, the transferred EV gDNA has pathophysiological significance, as a BCR/ABL hybrid gene involved in the pathogenesis of leukemia could be transferred from leukemia cells to human embryonic kidney cells or neutrophils through EVs [61]. Another study showed that the human *H-ras* oncogene in rat epithelial cells increased the production of EVs, which can be transiently transferred to recipient cells, increasing the recipient-cell proliferation [62]. In addition, mouse cardiomyocyte EV DNA transfer into target fibroblasts was proven, where DNA stains were observed in the fibroblast cytosol and nucleus. A total of 333 gene expressions were altered in the fibroblasts transfected with EVs [125].

Bacterial outer membrane vesicles (OMVs) can deliver DNA to other bacteria over long distances [55]. OMVs even have the ability to deliver DNA into other species of bacteria and eukaryotic host cells [56]. A study using bone marrow-derived mesenchymal stromal cells (MSCs) and foreign DNA of *Arabidopsis thaliana* demonstrated that EVs were capable of HGT between MSCs [126]. Furthermore, in patients with breast cancer, cancer-associated fibroblasts (CAFs) package mtDNA into EVs, which can contribute to the upregulation of mitochondrial genes required for oxidative phosphorylation when taken up by hormone therapy (HT)-resistant breast cancer cells. Specifically, the treatment of HT-naive cells or HT-treated metabolically dormant cells with CAF-derived EVs harboring mtDNA promoted an escape from metabolic quiescence and the development of drug resistance both in vitro and in vivo [127]. This shows that EV-mediated HGT was not limited to gDNA. The first study exploring the mechanism of HGT by exosomes suggests that HGT occurs in double-strand break repair through genome editing [128]. These results suggest that the EV-mediated lateral transfer of DNA between eukaryotic cells may result in aneuploidy and the accumulation of genetic factors, leading to tumor formation, which is a driving force behind mammalian genome evolution. Further investigation into the molecular mechanisms of EV-DNA-mediated gene transformation could open up several opportunities for cancer biology and therapeutics, and warn us of new risks for the leading-edge technology.

## 7. NGS Analysis Using Extracellular Vesicle-Derived DNA

In 2014, three independent studies profiled dsDNA isolated from EVs using NGS, verifying the possibility for clinical use for the first time [26,57,62]. Since then, several research groups have performed whole-genome sequencing (WGS), whole-exome sequencing (WES), and targeted NGS for the last seven years (Table 3). The revolutionary finding that EVs contain protected dsDNA opened new opportunities for their application as a more concentrated and better-preserved source of cancer-derived genomic material, as an alternative to cfDNA [129]. Moreover, EVs contain gDNA originating from long dsDNA, and NGS analysis is possible without the need for deep sequencing or barcoding NGS [41].

### 7.1. WGS and WES

Cai J et al. [61], using WGS, observed a minimum of 16,434 gDNA fragments in EVs from the plasma of healthy humans. Kahlert’s team [57] used exosome-derived DNA (exoDNA) from the serum of patients with pancreatic cancer and tumor DNA to map approximately 96% of the human genome using NGS. The properly paired percentage read ~92% between tumor gDNA and exosome gDNA. Their analysis showed that the DNA found in exosomes isolated from the serum was uniform across all chromosomes [57]. In another study, WGS and comparative genomic hybridization analysis with the exoDNA of murine melanoma cells revealed the entire genome coverage of exoDNA without bias. In addition, no specific fragments were enriched or depleted in the exoDNA pool compared to gDNA [26]. Lee et al. [62] investigated the copy number variation (CNV) of human *H-ras-*transformed rat intestinal epithelial cell EV-associated DNA. While the CNV research suggested an increase in the contribution of certain loci, they did not detect any genomic regions that would be selectively included in EVs. Indeed, over 90% of the cellular genome has been found in EVs [62]. A study with PC12 cells demonstrated that exoDNA covered 98% of the single nucleotide polymorphisms (SNPs) of the parent cell with WGS. They examined driver susceptibility gene mutations in exoDNA and found that the concordance rates of mutations in the exosome and tumor tissue DNA in patients with pheochromocytoma or paraganglioma were as high as 98–100% [131]. Interestingly, WGS analysis using exoDNA from the plasma and ascites of patients with ovarian cancer identified gene mutations related to DNA repair, and revealed a similarity to the primary tumor in CNV only with ascite exosomes, but not plasma exosomes [72]. Several studies have demonstrated the potential of EV DNA for WGS, and have shown that EV DNA represents the whole genome of parental cells [63,78,126,130]. A study used exoDNA from the pleural effusion and plasma of patients with pancreatic biliary cancer for analyzing the CNV, SNV, gene fusions, and mutational signature using WGS and WES. In addition, exosome RNA (exoRNA) was used for transcriptome sequencing, and WES using exoDNA covered 95–99% of the target regions. In this case, exoDNA was used for identifying mutant *KRAS* through WGS, WES, and transcriptome sequencing of pancreatic ductal adenocarcinoma [40]. Another study used large EV DNA isolated from the human prostate cancer cell PC3 to identify genomic rearrangements [35]. While CNV was not identified in normal blood samples, urinary cfDNA and exoDNA samples presented a similar pattern of CNV with tumor samples of urothelial bladder carcinoma [70]. In addition, fetal trisomy and single gene disease were identified by EV DNA in the maternal plasma. This study demonstrated that the GC content of the plasma EV DNA was 1.2 times higher than that of cfDNA. Moreover, mtDNA was detected in EVs using NGS, and the read percentage of mitochondrial EV DNA was, on an average, 2.2 times higher than that of cfDNA [132].

However, in a contradictory study, the EV DNA results were highly variable between patients with very limited overlapping regions, even when the EV DNA CNV profiles were compared to the CNV profile of formalin-fixed paraffin-embedded (FFPE)-derived DNA [134]. This difference could be due to the difference in DNA degradation during storage in FFPE tissue blocks [138,139]. In another study, only one case presented a lower sensitivity to tumor driver CNVs of breast cancer with EV DNA compared with ctDNA [133].

With neuroblastoma, a comparison of exoDNA and the corresponding tumor DNA using WES showed a higher overall number of somatic single nucleotide variants (SNVs). The difference in SNVs could arise from the different origins of exoDNA and tumor DNA, suggesting spatial genetic heterogeneity. In addition, the median tumor mutation burden (TMB) of exoDNA and tumor DNA was calculated. The higher TMB value observed in the exoDNA was probably because there were more exoDNA somatic SNVs than tumor DNA SNVs [135].

### 7.2. Targeted NGS

Many studies have tested the compatibility of EV DNA from different types of body fluids in various diseases for targeted NGS (Table 3). Evaluation of BALF EV DNA for the detection and quantification of mutations comparably identified lung adenocarcinomas with tissue DNA using targeted NGS. The DNA yield from BALF EV has been demonstrated to be 100 times higher than that from tissue samples. The median depth of coverage, median sequencing uniformity, and tumor purity were higher in the DNA from the tissue than in the BALF EV DNA. However, the estimated library size was not significantly different between the two samples, and the median DNA fragment length was slightly longer in the BALF EV DNA than in the tissue DNA. *EGFR* variants were the most commonly detected alterations, totaling 580 alterations in 175 genes. Furthermore, the overall mutation concordance between the two samples was high (81%) for clinically significant mutations. In addition, the TMB of the BALF EV DNA was correlated with that of the tissue DNA [41].

A study performed on urine samples of patients with urothelial bladder carcinoma identified all 17 somatic mutations by analyzing cfDNA and exoDNA [70]. The compatibility was also demonstrated in a glioblastoma study comparing DNA from EVs, cells, and matching tissues; the study determined the variant allele frequencies (VAFs) to be similar [106].

Several studies have used plasma-derived EV DNA for NGS analyses. One study explored the plasma of patients with advanced cancer (colorectal cancer, melanoma, and NSCLC) to identify common hotspot mutations such as *BRAF*, *EGFR*, and *KRAS* to yield a very good overall sensitivity (95%) with exoDNA and exoRNA compared with the standard testing of archival FFPE samples obtained from the tumor tissue. This high sensitivity of plasma exoDNA and exoRNA was similar to that of simultaneously tested plasma cfDNA with ddPCR and BEAMing (92% and 97%, respectively) [83].

For tumor exosome enrichment, using the plasma of pancreatic ductal adenocarcinoma (PDAC) patients, surface exosomal proteins were profiled to identify PDAC-specific biomarkers using proteomic analysis, which led to an augmentation of mutant genomic equivalents that were suitable for subsequent NGS using a molecular barcoding approach [136]. In another plasma study of acute myeloid leukemia patients, EV DNA analyzed for leukemia-specific mutations using NGS mirrored the leukemia-specific mutations found in the gDNA obtained from the primary leukemia cells in most cases [137].

These studies demonstrated that EV DNA in patients with cancer can be a reliable source for targeted NGS for the identification of genetic alterations using diagnostic values with a high clinical feasibility and utility. For clinically reliable and suitable NGS analysis in the future, standardization and clinical verification are necessary.

## 8. Challenges of Studying Extracellular Vesicles

The heterogeneity of EVs and the presence of non-vesicular extracellular nanoparticles in fluid samples pose major obstacles to our understanding of the composition and functional properties of the secreted EV components [74,140]. In particular, in EV DNA research, the use of DNase to remove outer DNA and non-vesicular extracellular nanoparticles, as well as other appropriate separation methods, should be considered. EVs are most often categorized according to their size. Exosomes are the smallest vesicles with sizes ranging between 30 and 150 nm, microvesicles typically range between 0.1 and 1 μm, and apoptotic bodies tend to have larger diameters of 50–5000 nm [29]. However, some microvesicles are smaller than 100 nm, such as the recently identified arrestin domain-containing protein 1-mediated microvesicles, which are relatively on the smaller side, 40–100 nm [141,142]. Therefore, most isolation protocols are based on differences in size and buoyant density, such as gradient centrifugation, sediment centrifugation, ultracentrifugation, and size exclusion chromatography. Unfortunately, these approaches cannot differentiate the population of EVs with diameters ranging between 50 and 200 nm, meaning that a pure population of exosomes, small apoptotic bodies, or microvesicles cannot be obtained using current methods. Consequently, most analyses are performed on EVs as a whole rather than on a pure exosomal population. Most studies use differential ultracentrifugation for the isolation of EVs, and some still prefer to use the term exosome. However, pure exosomes cannot be obtained by ultracentrifugation alone, and accordingly, it would be more appropriate to call them small EVs. To solve this problem, distinct surface markers can distinguish exosomes from other types of EVs [143].

Tetraspanins, CD63, CD81, and CD9 are most often used as specific markers for identifying exosomes; however, they are not a definite indication, as these proteins are abundantly expressed on the cell surface, including other types of EVs that are generated by budding from the plasma membrane [144,145]. Moreover, these markers of exosomes are not present in all cases; alternatively, some of the presumed components of exosomes were absent in the exosomes expressing these markers [74,146,147]. Therefore, a single marker cannot be used to identify all EVs within a population. Moreover, there is no method for isolating only a pure population of exosomes or microvesicles without traces of complex mixtures of EVs and other components from bio-fluids [29,148].

A method capable of isolating pure exosomes or microvesicles needs to be developed. Although limitations to absolute differentiation between these types of EVs still exist, EVs isolated according to their size could be categorized as small or large EVs, or those isolated by the difference in their densities could be called either high-density or low-density EVs. Additional tests of certain protein markers and the identification of isolated EVs as marker-positive EVs, such as CD63 + EV, could be supplemented for supporting the categorization of EVs [31]. In summary, one should always clarify the method of EV isolation and define the characteristics before choosing the right nomenclature for isolated EVs.

## 9. Discussion

EVs are nanoparticles of 30–1000 nm in size, and are found almost everywhere in bodily fluids [115]. They are shed by various types of cells, but are most abundantly released by cancer cells and carry not only RNAs and proteins, but also DNA, including ssDNA and dsDNA, which originate from the gDNA and mtDNA of the parent cell [65]. Several recent studies have discovered that EVs hold dsDNA, including oncogenic mutant DNA studies performed with EVs derived from the plasma [95], pleural effusion [10], and BALF [98] of patients with cancer. These studies demonstrated that EV DNA presents a strong potential as a biomarker.

The application of NGS for clinical use has led to a new era of precision medicine powered by targeted therapies for cancer. Especially in lung cancer, various targeted therapies have been developed, ranging from EGFR-TKIs and BRAF, ALK, ROS, RET, MET, TRK1, and HER2 inhibitors [149], to more recent developments in KRAS inhibitors [150]. For immune therapy, assessing TMB using NGS has received attention as a biomarker [151]. In dealing with patient samples, separately identifying the biomarkers and diagnosis would consume time and resources; however, using NGS would save specimens and time [152,153].

Unfortunately, in several cancers, small biopsy samples have become an obstacle for NGS. In reality, 30% of NSCLC patients face the problem of insufficient biopsy samples [154,155]. In addition to the small biopsy sample problem, other challenges faced in NGS are tumor heterogeneity [156] and artificial mutation of the fundamental FFPE tissues [157]. These shortcomings can be overcome through active clinical study of liquid NGS using plasma ctDNA [158,159], which is clinically applicable as liquid NGS lung cancer panels have become commercially available [160]. The biggest advantage of using plasma ctDNA is its easy access; however, its fundamental limitation is its instability, with a half-life of approximately 2 h [161], which indicates that it does not live up to the high expectations. In fact, liquid biopsy currently plays a supporting role in cases where tissue biopsy fails or when the location of a tumor renders it difficult to remove a tissue sample. Larger size genomic dsDNA within the EVs of cancers, including human glioblastoma, prostate cancer, ovarian cancer, and lung cancer, has been verified, and it was found in the serum, plasma, pleural effusion, urine, ascites, and BALF of patients with cancer. These studies verified that WGS, WES, and targeted NGS analysis using EV DNA are all possible (Table 3). Furthermore, targeted NGS using BALF EV DNA showed that NGS was possible without the molecular barcoding approach, and the results were highly consistent with somatic mutations in the tumor DNA [41]. The liquid biopsy method is moving away from conventional PCR and expanding into NGS; with the use of EV DNA, it could expand much further. In the liquid biopsy field, NGS using EV DNA will become clinically useful in cancer diagnosis, companion diagnostics, and prognosis in the near future.

In addition, the methylation of EV DNA from the serum, plasma, saliva, and gastric juice has been studied (Table 1), and similarities between gDNA and EV DNA results have been verified, which suggests that the methylation analysis of EV DNA can serve as a useful biomarker for the detection of various diseases, especially cancer. However, as some results lead to doubts about using EV DNA for methylation analysis [71,107], the EV DNA source, sample collection, DNA extraction methods, and the disease appear to affect the detection of methylation.

Although most current research on EV DNA has focused on cancer, non-cancer studies also present the potential for exploration. In particular, BALF can be a useful source for identifying the biomarkers of diseases, including idiopathic pulmonary fibrosis (IPF) and chronic pulmonary obstructive disease (COPD) [162].

EV DNA is involved in intercellular communication, pathological communication of diseases, and genomic evolution. For example, in cancer, oncogenes of donor cells can be transformed into recipient cells by apoptotic bodies or the BCR/ABL hybrid gene in EVs, and then expressed on recipient cells. Horizontal plant DNA transfer in eukaryotic cells is also mediated by EVs. Transfer from bacteria to eukaryotes by OMV-derived DNA has been identified in the nuclear fraction of epithelial cells. Furthermore, cells that become resistant to therapy can transfer mtDNA by EVs to other non-resistant cells and exhibit therapy resistance both in vivo and in vitro (Table 2). This transfer of EV DNAs suggests the potential of EVs as carrier vesicles and for other therapeutic uses.

Currently, most research and clinical studies on EV DNA have focused on discovering biomarkers for liquid biopsy and its horizontal gene transferability. The biggest problem for the clinical application of EV DNA is the lack of basic research and characterization. To advance our understanding of EV DNA, we must establish the (1) intracellular processes involved in the loading mechanism of DNA onto extracellular vesicles, (2) specific cellular signals that load DNA onto EVs, (3) the mechanism of DNA transfer to recipient cells via EV, and (4) an EV-DNA-specific optimized isolation and analysis method for removing non-vesicular extracellular nanoparticles. Once a full understanding of EV DNA has been established, it would become valuable in many aspects as a clinical tool for diverse functions.

## 10. Conclusions

While there is a wealth of research examining EVs and EV DNAs, our understanding of the basics and definite characterization remains low. As the benefit of utilizing EV DNA for liquid biopsy is unquestionable, further research is required in establishing a reliable method of EV and EV DNA purification, exploring diseases other than cancer, and performing clinical research. In addition, verification of horizontal EV DNA transfer suggests the possibility of EVs as carrier vesicles and for other therapeutic uses, where a rigorous study of EV DNA loading and transfer mechanism is needed.

## Figures and Tables

**Figure 1 cancers-13-03827-f001:**
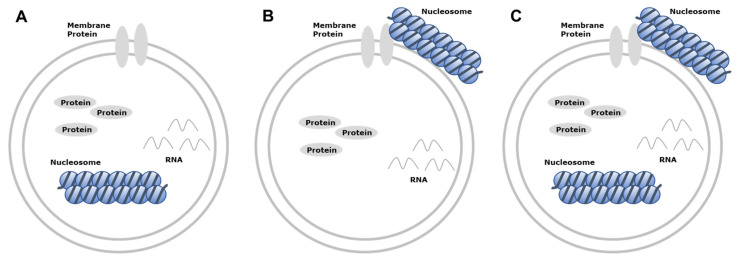
Characterization of DNA-loaded EVs. (**A**) DNA can be enclosed within EVs, (**B**) attached to the outer surface of EV, or (**C**) enclosed and attached to the outer surface. EV—extracellular vesicle.

**Figure 2 cancers-13-03827-f002:**
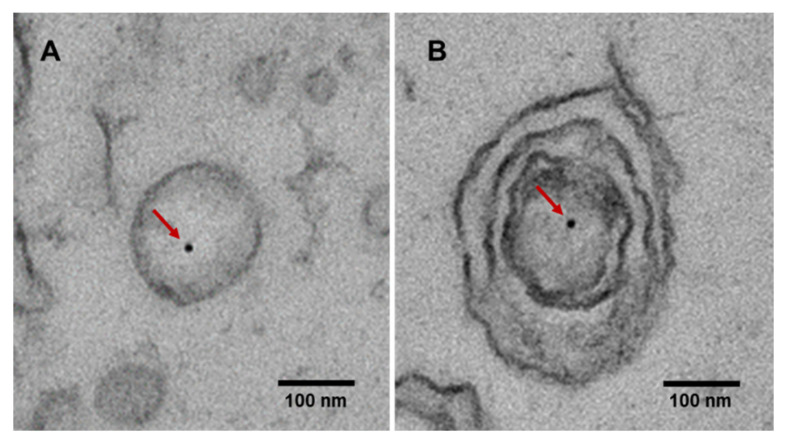
Detection of dsDNA in BALF EVs using immuno-EM. (**A**) Single vesicle and (**B**) multilayer vesicle. The solid black dots indicate DNA (indicated by red arrows). dsDNA—double-stranded DNA; BALF—bronchoalveolar lavage fluid; EV—extracellular vesicle; EM—electron microscopy.

**Figure 3 cancers-13-03827-f003:**
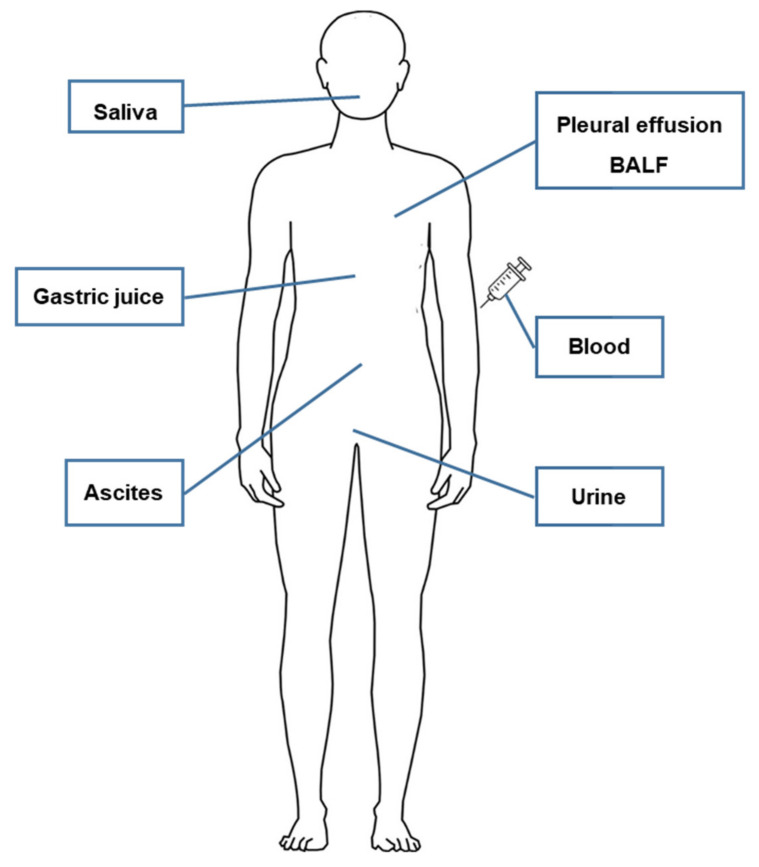
Body fluids as a source of DNA derived from EV. EV—extracellular vesicle; BALF—bronchoalveolar lavage fluid.

**Figure 4 cancers-13-03827-f004:**
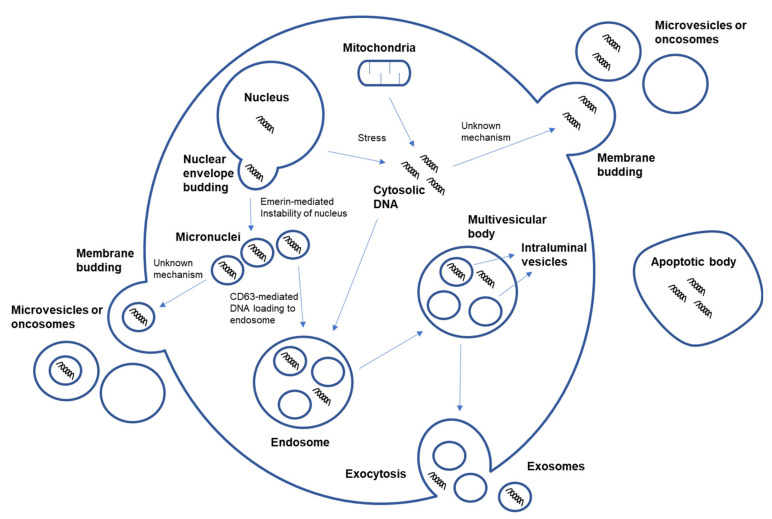
DNA loading into EVs via various mechanisms. DNA can be loaded into EVs through a CD63-mediated DNA shuttle or emerin-mediated nucleus instability and shedding. Other possible unknown mechanisms could include the loading of cytosolic DNA that originated from the nucleus and mitochondria by oxidative stress. EV—extracellular vesicle.

**Table 1 cancers-13-03827-t001:** Summary of methylation analysis performed with EV DNA.

Source of EV (Sample Size)	EV Type (Size)	Isolation Method of EV	Methylation Analysis (Gene)	Reference
Murine melanoma cell B16-F10	Exosome	Differential ultracentrifugation and filtration	Dot blot analysis	[26]
Gastric juice from patients with gastric cancer (*n* = 20)	Exosome (30–100 nm)	ExoQuick-TC	Bisulfite pyrosequencing (*SOX17*)	[73]
Gastric juice, gastric cancer (*n* = 20), and non-gastric cancer (*n* = 10)	Exosome	ExoQuick-TC	Bisulfite pyrosequencing (*BARHL2*)	[102]
Serum; normal (*n* = 7), breast cancer (*n* = 5), and melanoma (*n* = 4)	EV (30–250 nm)	Total Exosome isolation reagent	ELISA-based global DNA methylation analysis and microelectrode device	[103]
Saliva from patients with gingivitis (*n* = 7) and healthy individuals (*n* = 5)	Small EV (<200 nm)	Differential ultracentrifugation and filtration or size exclusion chromatography	Quantitative methylation-specific PCR (*IL−6*, *TNF-α*, *IL−1β*, *IL−8*, and *IL−10*)	[71]
Plasma from patients with prostate cancer (*n* = 62) and healthy individuals (*n* = 10)	Exosome	exoRNeasy Maxi kit	Real-time methylation-specific PCR (*GSTP1*, *RASSF1A*, and *SLFN11*)	[104]
Plasma from patients with lymphoma (*n* = 21) and healthy individuals (*n* = 21)	Exosome (40–120 nm)	Differential ultracentrifugation and filtration	Methylation-specific PCR (*CDKN1A*, *CDKN1B*, *CDKN2A*, and *CDKN2B*) or dot blot analysis	[105]
Glioblastoma stem-like cell (*n* = 8)	EV	Differential ultracentrifugation and filtration or size exclusion chromatography	Infinium methylation EPIC arrays	[106]
Saliva from patients with gingivitis (*n* = 7) and periodontitis (*n* = 8) and healthy individuals (*n* = 7)	Small EV (<200 nm)	Size exclusion chromatography	Global DNA methylation assay	[107]

EV—extracellular vesicle; EV DNA—EV-derived DNA.

**Table 2 cancers-13-03827-t002:** Research on the transfer of EV DNA.

Donor Cells	Recipient Cells	EV Type (Size)	Isolation Method of EV	Method of EV Transfer	Effect of EV DNA Transfer	Reference
EBV-carrying transformed lymphoid cells	Human fibroblast, macrophage, or bovine aortic endothelial cell	Apoptotic body		Co-culture	EBV-DNA and gDNA transfer to recipient cells via apoptotic bodies	[124]
*H-ras* and *c-myc* transformed rat embryonic fibroblasts	Mouse embryonic fibroblast	Apoptotic body		Co-culture	Oncogenes of donor cell (*H-ras* and *c-myc)* cause transformation of the recipient cells via apoptotic bodies	[123]
Mouse cardiomyocytes HL-1	Fibroblasts NIH 3T3	Microvesicle	Differential ultracentrifugation	Incubation of fibroblasts with EV	EV DNA transfer to fibroblasts	[125]
AT_1_ receptor transfected HEK293 cells or VSMC or leukemia cells, K562	HEK293 or human neutrophil	EV (30–1000 nm)	Differential ultracentrifugation	Incubation of HEK293 or human neutrophils with EVs	AT_1_ receptor or *BCR/ABL* hybrid gene transfer via EV and expression on recipient cells	[61]
*H-ras* transformed rat epithelial cells, RAS-3	Rat fibroblasts RAT-1	EV (100 nm)	Differential ultracentrifugation and filtration	Incubation of fibroblasts with EVs	Transfer of oncogenes of donor cells to recipient cells via EV	[62]
*Arabidopsis thaliana*-plasmid-transduced hMSC	hMSC	Small EV (50–150 nm)	Differential ultracentrifugation	Incubation of hMSC with EVs	Horizontal plant DNA transfer to eukaryotic cells via EV	[126]
*Pseudomonas aeruginosa*	Human adenocarcinoma A549	OMV (~20 nm)	Differential ultracentrifugation, filtration, and density gradient ultracentrifugation	Incubation of lung epithelial cells with OMV	OMV-derived DNA is detected in the nuclear fraction of epithelial cell	[56]
Murine cancer-associated fibroblasts from xenograft	Hormonal therapy-naïve breast cancer cell	EV (~140 nm)	Differential ultracentrifugation	Injecting mCAF EV into tumor-bearing mouse or incubation of breast cancer cells with EV	Transfer of therapy resistance to therapy-sensitive cells via mtDNA from EV in vivo and in vitro	[127]

EV—extracellular vesicle; EV DNA—EV-derived DNA; EBV—Epstein-Barr virus; VSMC—vascular smooth muscle cells; hMSC—human mesenchymal stem cell; OMV—outer membrane vesicles; mCAF—murine cancer-associated fibroblasts; mtDNA—mitochondrial DNA.

**Table 3 cancers-13-03827-t003:** Summary of NGS analysis performed with EV DNA.

NGS Type	Source of EV (Sample Size)	EV Type (Size)	Isolation Method of EV	Mean Depth	Subjects of Comparison	Number of Targeted Genes	Reference
WGS	Plasma of healthy humans (*n* = 30)	EV	Differential ultracentrifugation				[61]
WGS	Serum of patients with pancreatic cancer (*n* = 2)	Exosome	Differential ultracentrifugation and filtration	4×	CNV of tumor DNA and exoDNA		[57]
WGS	Murine melanoma cells, B16-F10	Exosome	Differential ultracentrifugation and filtration		CNV of exoDNA		[26]
WGS	Human H-ras transformed rat intestinal epithelial cells, RAS-3	EV (100 nm)	Differential ultracentrifugation and filtration	~7×	CNV of EV DNA		[62]
WGS	Pleural effusion (*n* = 1) and plasma (*n* = 2) of patients with pancreaticobiliary cancer	Exosome	Differential ultracentrifugation and filtration	12–35×	CNV, SNV, gene fusions, and mutational signature of tumor DNA (tissue) and exoDNA		[40]
WGS	Human bone marrow-derived MSC (*n* = 2)	Small EV (50–150 nm)	Differential ultracentrifugation		CNV of EV DNA		[126]
WGS	Malaria parasite-infected human red blood cells	EV (50–350 nm)	Differential ultracentrifugation, filtration, and density gradient ultracentrifugation		CNV of malaria gDNA and EV DNA		[130]
WGS	Human fibroblasts, TIG-3	Exosome	Differential ultracentrifugation, filtration, and density gradient ultracentrifugation		RPKM of genomic DNA and EV DNA		[78]
WGS	Urine of patients with urothelial bladder carcinoma (*n* = 9)	Exosome	ExoQuick-TC	0.6×	CNV of tumor DNA and urinary DNA (cfDNA and exoDNA)		[70]
WGS	Serum of patients with pheochromocytoma and paragangliomas and rat cells, PC12	Exosome	Differential ultracentrifugation		SNP of tumor DNA and exoDNA		[131]
WGS	Human prostate cancer cells, PC3	Large EV (1.0–5.5 μm)	Differential ultracentrifugation and density gradient ultracentrifugation	~1.4×	CNV and genomic rearrangements of gDNA and EV DNA		[35]
WGS	Human erythroleukemic cells, TF-1, and mast cells, HMC-1	Small EV (~120 nm)	Differential ultracentrifugation and density gradient ultracentrifugation	9.25–15.88 ×	CNV of EV DNA		[63]
WGS	Plasma and ascites of patients with ovarian cancer (*n* = 3) and human ovarian cancer cells, OVCAR-5	Exosome	Differential ultracentrifugation and filtration	20×	CNV and SNV of tumor DNA (tissue) and exoDNA (plasma and ascites)		[72]
WGS	Maternal plasma (*n* = 20)	EV (30–50 nm)	ExoQuick	0.25×	cfDNA and EV DNA		[132]
WGS	Plasma of patients with breast cancer (*n* = 1, serial samples (X3))	EV (30–600 nm)	Differential ultracentrifugation	1×	CNV of tumor DNA (FFPE), ctDNA, and EV DNA		[133]
WGS	Plasma of patients with tongue base squamous cell carcinoma (*n* = 3) and cutaneous squamous cell carcinoma (*n* = 2)	EV (215 nm)	Differential ultracentrifugation	0.5–1×	CNV of tumor DNA (FFPE) and EV DNA		[134]
WES	Pleural effusion (*n* = 1) and plasma (*n* = 2) of patients with pancreaticobiliary cancer	Exosome	Differential ultracentrifugation and filtration	133–490×	SNV and mutational signature of tumor DNA (tissue) and exoDNA		[40]
WES	Plasma of patients with neuroblastoma (*n* = 19)	Exosome	Exo-RNeasy serum/plasma midi kit	110×	SNV and TMB of tumor DNA (FFPE) and exoDNA (plasma)		[135]
Targeted NGS	Plasma of patients with advanced cancers (*n* = 43)	Exosome	ExoLution Plus Isolation kit		SNV of tumor DNA (tissue) and exoNA	3	[83]
Targeted NGS	Urine of patients with urothelial bladder carcinoma (*n* = 9)	Exosome	ExoQuick-TC	102–4909×	SNV of tumor DNA (tissue), cfDNA, and exoDNA	9	[70]
Targeted NGS	Plasma of PDAC patients	Exosome	Differential ultracentrifugation and cancer-specific exosome capture		SNV of tumor DNA (tissue) and exoDNA	275	[136]
Targeted NGS	BALF of patients with lung adenocarcinoma (*n* = 20)	EV (207.0 ± 48.3 nm)	Differential ultracentrifugation	190–755×	SNV of tumor DNA (FFPE) and EV DNA	375	[41]
Targeted NGS	Plasma of patients with acute myeloid leukemia (*n* = 4)	EV (30–150 nm)	Differential ultracentrifugation		SNV of tumor DNA and EV DNA	54	[137]
Targeted NGS	Glioblastoma stem-like cells (*n* = 8)	EV	Differential ultracentrifugation and filtration or size exclusion chromatography		SNV of tumor DNA (tissue and cell) and exoDNA	47	[106]

NGS—next-generation sequencing; EV—extracellular vesicle; EV DNA—EV-derived DNA; WGS—whole-genome sequencing; CNV—copy number variation; exoDNA—exosome-derived DNA; SNV—single nucleotide variant; MSC—mesenchymal stromal cell; gDNA—genomic DNA; RPKM—reads per kilobase per million mapped reads; cfDNA—cell-free DNA; SNP—single nucleotide polymorphism; FFPE—formalin-fixed paraffin-embedded; WES—whole-exome sequencing; TMB—tumor mutation burden; PDAC—pancreatic ductal adenocarcinoma; BALF—bronchoalveolar lavage fluid.

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
