# Peer review of "Characteristics and Clinical Application of Extracellular Vesicle-Derived DNA"

_cancers, 2021, doi:10.3390/cancers13153827_

Round 1
Reviewer 1 Report
In this review, author summarized the functions of extracellular vesicle–derived DNA. The review appears to be quite comprehensive, however, more cartoons would be helpful in the understanding of the broad effects of interleukins. There are a few concerns that I would strongly recommend being addressed:
- In Figure 2, it should be BALF not BLAF.
- It would be ideal if the authors can also include some non-cancer studies, especially for the BALF part.
- Ch. 6 Mechanism of loading DNA to Extracellular vesicle has limited content, it should be extended or combine with Ch. 7 Transfer of Extracellular vesicle–derived DNA
- Many grammatical errors are identified. The entire manuscript will need to be carefully proof-read and edited by a professional language service.
Author Response
Responses to the comments
Reviewer 1
In this review, author summarized the functions of extracellular vesicle–derived DNA. The review appears to be quite comprehensive, however, more cartoons would be helpful in the understanding of the broad effects of interleukins. There are a few concerns that I would strongly recommend being addressed:
Reply: Thank you for your advice. As you have suggested, we have added two figures (figure 1 and figure 4) that could be helpful in understanding extracellular vesicles. I believe the interleukin you mentioned is a typo.
- In Figure 2, it should be BALF not BLAF.
Reply 1: We fixed this error.
- It would be ideal if the authors can also include some non-cancer studies, especially for the BALF part.
Reply 2: If you read the manuscript closely, it already includes non-cancer studies such as gingivitis, periodontitis, malaria, and fetal DNA from maternal plasma. Most of the EV-related studies are indeed focused on cancer as that is the trend for few reasons. One of the reasons, as mentioned in the review, is that cancer cells tend to release more EVs, which makes them an ideal target for diagnostic research. Therefore, most of the EV–derived DNA research is done in cancer, making reviewing non-cancer studies difficult. Unfortunately, we could not find any published data or studies that dealt with BALF EV–derived DNA in non-cancer studies. It would be interesting for researchers to explore non-cancer studies in near future, which sentiment we have added to the manuscript. (see Page 19, line 504-507).
- Ch. 6 Mechanism of loading DNA to Extracellular vesicle has limited content, it should be extended or combine with Ch. 7 Transfer of Extracellular vesicle–derived DNA
Reply 3: As you have suggested, Ch. 6 and Ch.7 were combined. (see Page 8-10, line 212-310).
- Many grammatical errors are identified. The entire manuscript will need to be carefully proof-read and edited by a professional language service.
Reply 4: The manuscript has been proof-read and edited by a professional language service to improve overall grammar and readability. The details of the edit are not marked as they did not change the contents or their meaning.
Reviewer 2 Report
The presented paper is the product of an extensive review of recent literature on EV-associated DNA, and includes several and well organized examples. The paper is generally well written and easy to understand, and all results are clearly presented.
Despite the overall good quality of the paper, I have several concerns regarding the presence of DNA (especially dsDNA) in EVs. In fact its presence or not is controversial and still a matter of debate for the scientific community.
Moreover some data described in this review appears to be contradictory:
- Page 3, lines 115-116: “The size of dsDNA found in EVs range from ~100 bp to ~20 kbp, which can represent the entire genome and reflect the mutational status of tumor parental cells”
- Starting from this sentence, and considering that the distance between two base pairs in dsDNA is 0.34 nm, EV dsDNA should be long from 34 nm to 6.8 µm. Since dsDNA is a rigid macromolecule I do not figure out how an EV, which ranges from 50 nm to 5 µm in diameter (page 16 line 414), can accommodate DNA molecules longer than its diameter.
- For the same reason I do not understand how bacterial outer membrane vesicles (OMV) can contain dsDNA of at least 100 kbp.
Other topics of discussion about DNA presence in EVs are if DNA is actually EV-related or if it comes as a contamination from separation processes, and the actual location of DNA in EVs (inside as a cargo biomolecule or associated with the outer membrane by means of DNA binding proteins, as described by Galbiati et al. in this paper: Pharmaceuticals 2021, 14(2), 128; https://doi.org/10.3390/ph14020128).
The paper does not provide answer to these questions since in almost all cited examples EVs are purified via ultracentrifugation or polymer induced precipitation, which are known to provide EV samples contaminated by biopolymers (including DNA).
This fact would also explain why in many examples there is no difference between EV DNA and cfDNA, despite there is a claim for DNA protection from degradation inside EVs. Moreover with respect to these evidencies, why someone should isolate EVs when it is possible to obtain the same results analyzing cfDNA?
I do think that the presence and location of DNA inside the EVs still needs to be clearly demonstrated, as optimal separation protocols and analysis techniques still need to be developed to obtain an unequivocal response. In my opinion, this should clearly stated at the beginning of this review because otherwise it might be misleading and does not reflect the actual knowledge about EV-related DNA.
Author Response
Responses to the comments
Reviewer 2
he presented paper is the product of an extensive review of recent literature on EV-associated DNA, and includes several and well organized examples. The paper is generally well written and easy to understand, and all results are clearly presented.
Reply: Thank you for your advice.
Despite the overall good quality of the paper, I have several concerns regarding the presence of DNA (especially dsDNA) in EVs. In fact its presence or not is controversial and still a matter of debate for the scientific community.
Moreover some data described in this review appears to be contradictory:
- Page 3, lines 115-116: “The size of dsDNA found in EVs range from ~100 bp to ~20 kbp, which can represent the entire genome and reflect the mutational status of tumor parental cells”
Starting from this sentence, and considering that the distance between two base pairs in dsDNA is 0.34 nm, EV dsDNA should be long from 34 nm to 6.8 µm. Since dsDNA is a rigid macromolecule I do not figure out how an EV, which ranges from 50 nm to 5 µm in diameter (page 16 line 414), can accommodate DNA molecules longer than its diameter.
For the same reason I do not understand how bacterial outer membrane vesicles (OMV) can contain dsDNA of at least 100 kbp.
Reply: I understand your concern, but please keep in mind that this is a review paper and we did try to address both sides of controversies and include recent studies, which could have made the review seem contradictory at some points. By reviewing many studies regarding EV–derived DNA, we believe the debate arises from the heterogeneity and mixed definition of EVs. The reviewer’s comment regarding the size of the DNA and EVs could be correct if we are just considering the naked DNAs. However, big-sized DNAs existing in nucleosome or supercoiled form that could be packaged into EVs such as oncosome. This is demonstrated in studies by Yokoi A. (Sci Adv 2019, 5(11), eaax8849; https://doi.org/10.1126/sciadv.aax8849) and Lázaro-Ibáñez E. (J Extracell Vesicles 2019, 8(1), 1656993; https://doi.org/10.1080/20013078.2019.1656993), where histone and nucleus proteins are found with EV–derived DNA. Also, to clarify, the manuscript already mentions that packaging of dsDNA in small EVs is unlikely since the size of the nucleosome is only about 11nm. (see Page 3, line 108-110) Moreover, we made few additions following reviewer's comment. (see Page 3, line 106-108)
Other topics of discussion about DNA presence in EVs are if DNA is actually EV-related or if it comes as a contamination from separation processes, and the actual location of DNA in EVs (inside as a cargo biomolecule or associated with the outer membrane by means of DNA binding proteins, as described by Galbiati et al. in this paper: Pharmaceuticals 2021, 14(2), 128; https://doi.org/10.3390/ph14020128).
The paper does not provide answer to these questions since in almost all cited examples EVs are purified via ultracentrifugation or polymer induced precipitation, which are known to provide EV samples contaminated by biopolymers (including DNA).
Reply: As you pointed out, contamination of other DNAs in EV–derived DNA research is a problem. To resolve this problem, most of the studies treated DNase to the EV sample and used density gradient centrifugation to separate EVs.
This fact would also explain why in many examples there is no difference between EV DNA and cfDNA, despite there is a claim for DNA protection from degradation inside EVs. Moreover with respect to these evidencies, why someone should isolate EVs when it is possible to obtain the same results analyzing cfDNA?
Reply: While some studies only showed the same results between EV–derived DNA and cfDNA, other studies have demonstrated higher sensitivity with EV–derived DNA compared to cfDNA, demonstrating the need for using EV DNAs (Ann Oncol 2018, 29 (12), 2379; https://doi.org/10.1093/annonc/mdy458, Ann Oncol 2017, 28(4), 741; https://doi.org/10.1093/annonc/mdx004, Mol Cancer 2018, 17(1), 15; https://doi.org/10.1186/s12943-018-0772-6, Cancers 2020, 12(10), 2822; https://doi.org/10.3390/cancers12102822 and Gastroenterology 2019, 156(1), 108; https://doi.org/10.1053/j.gastro.2018.09.022). In addition, using EV–derived DNA enabled NGS without the costly deep sequencing and molecular barcoding, which is one of the reasons for isolating EVs (Transl Lung Cancer Res 2021, 10(1), 104; https://doi.org/10.21037/tlcr-20-888). Moreover, as mentioned in this review, superior results can be obtained by analyzing EV–derived DNA and cfDNA at the same time. (see Page 3, line 123-126)
I do think that the presence and location of DNA inside the EVs still needs to be clearly demonstrated, as optimal separation protocols and analysis techniques still need to be developed to obtain an unequivocal response. In my opinion, this should clearly stated at the beginning of this review because otherwise it might be misleading and does not reflect the actual knowledge about EV-related DNA.
Reply: We have edited the manuscript as you have suggested. (see Page 18, line 423-425 and Page 20, line 523-524)
Round 2
Reviewer 1 Report
The authors made all corrections needed.
Reviewer 2 Report
The authors' answers are satisfying and completely address my previous concerns.
I do recommend the current review for publication.